# Influence of Retinol Dermal Delivery Formulation on Its Stability Characteristics

**DOI:** 10.3390/gels11120935

**Published:** 2025-11-21

**Authors:** Ioana Lavinia Lixandru Matei, Bogdan Alexandru Sava, Codruta Sarosi, Cristina Maria Dușescu-Vasile, Andreea Iuliana Ionescu, Abeer Baioun, Marian Băjan, Gheorghe Brănoiu, Daniela Roxana Popovici, Andra-Ioana Stănică, Dorin Bomboș

**Affiliations:** 1Botanical SRL, 7 Trandafirilor St., 107059 Tătărani, Romania; 2Doctoral School of Applied Chemistry and Materials Science, University Politehnica Bucharest–UPB, 313 Splaiul Independenței, 060042 Bucharest, Romania; 3National Institute of Laser, Plasma and Radiation Physics-INFLPR, 409 Atomistilor St., 077125 Măgurele, Romania; 4“Raluca Ripan” Institute of Research in Chemistry, Babes–Bolyai University, 1 Mihail Kogălniceanu St., 400084 Cluj-Napoca, Romania; liana.sarosi@ubbcluj.ro; 5Petroleum Refining Engineering and Environmental Protection Department, Petroleum-Gas University of Ploiesti, 39 Bucharest Boulevard, 100515 Ploiești, Romania; marian.bajan@upg-ploiesti.ro; 6Chemistry Department, Faculty of Science, University of Damascus, Damascus P.O. Box 30621, Syria; 7Petroleum Geology and Reservoir Engineering Department, Petroleum-Gas University of Ploiesti, 39 Bucharest Boulevard, 100515 Ploiești, Romania; gheorghe.branoiu@upg-ploiesti.ro; 8Chemistry Department, Petroleum-Gas University of Ploiesti, 39 Bucharest Boulevard, 100515 Ploiești, Romania; dana_p@upg-ploiesti.ro; 9Technological Highschool “Toma Socolescu”, Gheorghe Grigore Cantacuzino St., 328, 100466 Ploiești, Romania

**Keywords:** retinol, hydrogel, emulsion, stability, bioglass

## Abstract

New cosmeceuticals formulas (direct emulsion, inverse emulsion and hydrogel), that synergistically combine bioglass with retinol, were prepared and characterized in order to attenuate the irritant potential of retinoids and prolong their therapeutic efficacy. The study evaluates the physicochemical, microbiological and stability characteristics of these formulations. Thus, TGA and DSC analyses revealed stronger interactions between water molecules and those of other organic compounds, with much more being observed in the case of emulsions than in the case of hydrogel, materialized by the delay in water evaporation. The stability of the three types of formulations has been evaluated in two ways: by determining the backscattering variation with the height of the container and analyzing the sample for 6 h and by counting the fraction of small droplets. Both methods demonstrated high stability in the three types of formulations.

## 1. Introduction

Skin aging is characterized by a gradual decline in both structural integrity and physiological function, influenced by intrinsic factors such as genetics and hormonal changes, as well as extrinsic factors like UV radiation, pollution, and lifestyle choices. The cosmeceuticals industry continuously seeks innovative solutions to effectively address visible signs of aging while promoting long-term skin health. Among the most extensively researched and clinically validated active ingredients is retinol (vitamin A), renowned for its ability to accelerate cell turnover, stimulate collagen production, and diminish the appearance of fine lines and hyperpigmentation. However, its use is often constrained by potential irritation, instability, and diminished efficacy in sensitive skin types. Recent advancements in biomaterials have introduced bioglass, a bioactive silica-based material traditionally employed in regenerative medicine, as a promising candidate for dermatological applications. Bioglass exhibits notable anti-inflammatory, antimicrobial, and wound-healing properties, along with an exceptional capacity to promote cellular regeneration and dermal remodeling. These characteristics position bioglass as a potential enhancer of skin barrier function and as a delivery platform for active compounds.

This study presents the development and characterization of an innovative cosmeceutical formula that effectively combines bioglass with retinol. The aim is to reduce the irritative potential of retinoids while enhancing their therapeutic effectiveness. The research discusses how the inclusion of bioglass can modulate the release profile of retinol, improve skin tolerance and provide additional regenerative benefits. The study evaluates the physicochemical properties, in vitro biocompatibility, and preliminary dermatological effects of this groundbreaking formulation. The evolution and improvement of hybrid gel systems—especially bigels (a mixture of hydrogels and oleogels)—gained significant attention in the food science field. A noteworthy contribution to this field is the study by Zhou et al. [1], who investigates the preparation of a two-phase gel system using gelatin-based hydrogels and oleogels made from beeswax and rice bran wax. This study explores the eutectic phase behavior of beeswax and rice bran wax in soybean oil, offering essential insights into the thermal and structural compatibility of these natural waxes within oleogel systems. The results of the research showed the specific behavior of oleogels (both melting and firmness), which is crucial for their integration into complex food applications. In a complementary effort, Hu et al. developed a novel margarine formulation by combining starch-based hydrogels with edible wax oleogels. The researchers successfully substituted conventional fats with this bigel structure, resulting in a product that boasts comparable spreadibility and enhanced health attributes. This work not only promotes cleaner-label fat alternatives but also emphasizes the role of starch hydrogels in stabilizing bigel systems [2].

Additionally, Xie et al. explored the functional versatility of bigels by examining how different oleogel/hydrogel ratios and emulsifier selections influence the structural and digestive characteristics of 3D printed bigels intended as carriers for polyphenols, including quercetin and catechin. Their research highlights the potential of bigel-based systems for controlled nutrient delivery, particularly through emerging technologies such as 3D food printing [3]. An important innovation in the field is the use of Pickering emulsions, which are stabilized by solid particles rather than traditional surfactants. Yuange Li and co. developed agar microspheres as particulate stabilizers, presenting an eco-friendly alternative that offers improved stability and compatibility with skin. These structured emulsions represent a promising approach for encapsulating and delivering active ingredients in cosmetic products [4].

The tactile properties of topical formulations play a significant role in consumer acceptance. Ali et al. examined the frictional behavior of creams and emulsions on both excised skin and synthetic membranes utilizing ForceBoard™. Their findings offer valuable insights into the sensory perception of these products, enabling formulators to better customize cream textures and skin feel [5].

In conjunction with advancements in formulation science, researchers are increasingly investigating the potential of plant extracts for their antioxidant and anti-aging properties. A comprehensive review by Xie et al. examines the efficacy of plant-derived compounds in mitigating oxidative stress and enhancing skin health. This research underscores the viability of botanical-based cosmeceuticals as a promising avenue for skin care interventions [6].

Nanotechnology is revolutionizing the field of cosmetic science by enabling the precise delivery of active ingredients and enhancing their bioavailability. Dubey et al. emphasized the incorporation of nanocarriers—such as liposomes, nanoemulsions, and dendrimers—into cutting-edge skincare formulations, which significantly improves therapeutic results [7]. A notable example of this technique is the encapsulation of retinol within water-soluble chitosan nanoparticles, as demonstrated by Kim et al. This method offers sustained release, boosts skin penetration, and minimizes irritation [8].

The significance of hyaluronic acid (HA) has garnered increasing attention due to its hydrating and regenerative properties. Al-Qadi et al. explored the interaction between HA and chitosan-based nanoparticles, highlighting its dual role as both a bioactive agent and an encapsulation enhancer [9]. In addition, Sakulwech et al. developed nanoparticles that combine quaternized cyclodextrin-grafted chitosan with HA, demonstrating promising skin compatibility and controlled release properties for cosmetic applications [10]. Further advancing these innovations, Liverani et al. incorporated bioactive glass nanoparticles into electrospun PCL/chitosan fibers, showcasing their potential in skin regeneration and wound healing, thereby bridging the gap between cosmetics and biomedical science [11]. From a regulatory and sustainability perspective, the investigation of natural preservatives is also underway. Kerdudo et al. presented a case study on the development of a natural preservative, underscoring the industry’s commitment to clean-label formulations [12].

Bioactive glasses, especially those with composition type 58S, have garnered considerable interest in biomedical applications due to their ability to bond with bone and promote tissue regeneration. Polyethylene glycol (PEG), a biocompatible polymer, is frequently incorporated into bioactive glass systems to modify their physical and microstructural properties. A prior study conducted by Ioana Lavinia Lixandru Matei et al. explored the impact of PEG 4000 on the structural, morphological, and textural characteristics of bioactive glasses of type 58S, aiming to optimize their performance for potential cosmetic applications [13].

Together, these investigations highlight a dynamic landscape where bioengineering, nanotechnology, and sustainability intersect to develop the next generation of cosmetic and skincare products.

This study explores the development and characterization of innovative cosmeceuticals formulations that synergistically combine bioglass with retinol. The objective is to reduce the irritant potential of retinoids while extending their therapeutic efficacy over time. To achieve this, three formulation variants were created, namely a direct emulsion, an inverse emulsion, and a hydrogel, each incorporating bioglass powder. This addition aims to modulate the release profile of retinol, enhance skin tolerance, and offer supplementary regenerative benefits. The research assesses the physicochemical, microbiological, and stability attributes of these formulations.

## 2. Results and Discussion

### 2.1. Thermal Properties

In Figure 1, the thermogravimetric analyses for the two emulsions are presented, and in Figure 2, thermogravimetric analyses are presented for the hydrogel. The hydrogel demonstrated a pronounced mass loss during the initial portion of the TGA curve, specifically within the temperature range of 40 °C to 130 °C, with a peak mass loss observed around 125 °C. This significant mass loss is primarily attributed to the evaporation of water. In the subsequent heating phase, spanning from 130 °C to 230 °C, the rate of mass loss diminished. This phenomenon is likely due to the evaporation of various volatile organic compounds, including triethanolamine (TEA). Beyond 230 °C, further mass losses were noted, albeit at a reduced rate, with no additional maxima detected in this temperature range. These losses can be ascribed to both the evaporation of organic compounds carried by the inert gas and the thermal degradation of non-volatile organic constituents such as hyaluronic acid, retinol, and the Carbopol Aqua SF-1 OS polymer. In comparison, the inverse emulsion displayed a lower mass loss in the initial segment of the TGA curve (40 °C to 130 °C) relative to the hydrogel, similarly attributed to water evaporation. During the subsequent heating phase, extending from 130 °C to approximately 330 °C, the mass losses were further decreased. These mass losses are likely a consequence of the evaporation of non-volatile organic compounds including TEA, lanolin, and various vegetable oils, facilitated by the carrier gas. At temperatures exceeding 330 °C, there was an increase in mass loss, with a notable peak at around 375 °C. The mechanism underlying these losses involves both the evaporation of less volatile organic compounds entrained by the carrier gas and their subsequent thermal decomposition.

The dry direct emulsion exhibited a lower mass loss compared to the inverse emulsion or hydrogel in the initial segment of the TGA curve (between 40 °C and approximately 220 °C). This loss is attributed to the evaporation of residual water and more volatile organic compounds. The first significant point of mass loss noted on the DTG curve occurred at around 260 °C and was relatively low in value. The principal maximum mass loss identified on the DTG curve was observed at approximately 380 °C, which is close to the temperature of the final maximum mass loss for the inverse emulsion. This mass loss can be attributed to both the evaporation of low-volatile organic compounds carried by the carrier gas and their subsequent thermal decomposition. In the case of the hydrogel, a single pronounced mass loss plateau is observed, as a result of the evaporation of water, which enters its composition in a proportion of 76% by mass. In relation to the two emulsions, the absence of a secondary mass loss interval indicates that the polymer used to stabilize the emulsion does not thermally decompose. At the same time, the temperature range of 180–600 °C indicates that the active substances, such as retinol and hyaluronic acid, are thermally stable.

The impact of the physicochemical characteristics of cosmetic creams on thermal transitions is elucidated through the Differential Scanning Calorimetry (DSC) thermogram, as depicted in Figure 3. The thermograms for the hydrogel and the two emulsion types exhibit a comparable profile. Notably, the initial segment of all three curves reveals an endothermic peak, characterized by a maximum occurring at temperatures exceeding 120 °C, attributed to the evaporation of water. The temperature at which this endothermic peak is observed is influenced by the nature of the interactions between water molecules and other constituents present in the formulation. The thermal stability of emulsions is contingent upon both the emulsion type and its compositional makeup. Specifically, in the case of direct emulsions, the interactions between water molecules (acting as the continuous phase) and other components are diminished, resulting in a peak temperature for water evaporation that is recorded at a point 3 °C lower than that of the hydrogel. Conversely, the hydrophilic polymer employed in the hydrogel formulation facilitates the retention of water molecules through hydrogen bonding and electrostatic interactions. In contrast, inverse emulsions display potentially greater stability due to their hydrophobic characteristics. For these indirect emulsions, the interactions of water molecules (in the discontinuous phase) with other components are intensified, needing a higher energy input for the migration of water molecules from the droplets into the surrounding environment. Consequently, the maximum temperature of the water evaporation peak is observed at a temperature 3 °C higher than that of the hydrogel.

Furthermore, the presence of minor endothermic peaks identified in both direct and reverse emulsions at elevated temperatures (approximately 380 °C) may indicate the thermal degradation of organic constituents, such as surfactants or vegetable oils incorporated within the emulsions. The discrepancy observed between direct emulsions (DE) and reverse emulsions (RE) can be ascribed to variations in the composition of the continuous and dispersed phases, thereby influencing the thermal stability of the organic compounds present within each formulation.

The different thermal behavior of the three formulations is due to both their different composition and their structure. Thus, the water loss highlighted by both TGA and DSC analysis occurs at relatively different temperature values, due to the different interactions between water molecules and the other constituents of these formulations.

### 2.2. FTIR Analysis

The results of the FTIR analysis, illustrated in Figure 4, reveal that the stretching vibrations observed at 3400 cm^−1^ are associated with the O-H bonds found in both water and triethanolamine (TEA). In the hydrogel, the shift in this band toward lower wavenumber values can be attributed to the incorporation of polyacrylic acid, which was used as a surfactant at a concentration of 6%. For all three emulsions analyzed, the vibrations at 1633 cm^−1^ are linked to the C=C bonds present in the retinol structure. The stretching vibrations recorded at 2926 cm^−1^ in the RE and DE are a result of C-H bonds found in the aliphatic chains of fatty acids derived from coconut and soybean oils. Additionally, the vibrations at 1747 cm^−1^ correspond to C=C bonds within the structure of unsaturated triglycerides, while those at 1149 cm^−1^ highlight the ester bonds typical of triglycerides. These signals are only highlighted in the case of the two emulsions, as the hydrogel does not contain coconut oil, soybean oil or lanolin. Lastly, vibrations below 1000 cm^−1^ indicate the presence of bioglass, characterized by Si-O bonds.

### 2.3. XRD Analysis

X-ray diffraction investigations reveal the amorphous nature of the two emulsions (Figure 5), with the diffraction spectra displaying a broad peak accompanied by a hump effect in the range of 15–25 degrees 2θ. This characteristic is typical of gels and oleogels and has been noted in several previous studies [3,14]. It is well established that various types of oils and fats, including those used in this study, such as palm oil, soybean oil, and lanolin, can exhibit crystalline phases, particularly in a solidified state or when mixed with other substances. Therefore, in conjunction with the broad peak observed in the range of 15–25 degrees 2θ, the XRD spectra illustrate several distinct diffraction peaks at d = 11.34 Å, d = 5.70 Å, d = 4.54 Å, d = 4.15 Å, and d = 3.83 Å, which correspond to 2θ angles of 7.70, 15.60, 19.50, 21.35, and 23.20 degrees.

The diffraction peaks observed at 2θ values of 7.70°, 15.60°, 19.50°, 21.35°, 23.20° and 25.26° correspond to characteristic peaks of the α and β’ polymorphic modifications of the fat molecules employed as raw materials for the emulsions (coconut oil, soybean oil, and lanolin). These peaks indicate that, at least in part, the initial fats were immobilized within the emulsion/oleogel in the form of fat crystals. This observation is supported by several previous studies, including those by Busakorn Mahisanunt et al. (2020) on coconut oil crystallization in the presence of tripalmitin and tristearin seed crystals with varying polymorphs [15,16] and Nguyen et al. (2019), who reviewed granular crystals in palm oil-based shortening/margarine [17]. Additionally, research by Sai Sateesh Sagiri et al. (2013) discussed lanolin-based organogels as matrices for topical drug delivery [18] and Ginsburg et al. (2024) examined the chemical transesterification of coconut and corn oils with various metal hydroxides to assess the chemical and physiochemical changes to the oils [19]. According to Dassanayake et al. (2009), the β′ polymorph exhibits excellent dispersibility, a smooth texture, and moderate melting characteristics due to its fine and uniform crystalline structure [20].

The diffraction spectra obtained from both direct and reverse emulsions encompass contributions from both crystalline and amorphous solids, revealing specific diffraction peaks at of 7.70°, 15.60°, 19.50°, 21.35°, and 25.26°, which are characteristic of the retinol structure. This observation corroborates the presence of retinol within the emulsions under investigation, as documented in previous studies [21,22,23]. The quantitatively similar water content in the two emulsions did not exert a significant influence on the intensity of the diffraction peaks associated with the fat/lanolin crystals. An in-depth analysis of the diffractograms indicates a decreased degree of crystallinity of the emulsions, a finding not attributable to the heightened water content observed in the emulsion gels. The enhanced intensity of peaks corresponding to the fat/lanolin crystals can be attributed to a greater degree of crystalline structure and the presence of long-chain molecular configurations, particularly during the formation of the β’ polymorph of lanolin/oleogel. These observations are consistent with FTIR analysis, confirming the presence of coconut or soybean oil in emulsions. For the reverse emulsion, the oil content is higher than in direct emulsion, the XRD pattern exhibiting more intense peaks. The similarity in the diffraction profiles of both the direct and indirect emulsions suggests comparable structural and compositional attributes, implying that the emulsifiers employed had a minimal impact on the crystalline architecture of the oleogels. Significantly, the diffraction spectrum for the inverse emulsion shows a more distinct peak intensity, likely due to the high water content. This enhances anisotropic growth, characterized by different growth rates in various directions. The intensity of the long-spacing diffraction modes is significantly lower when compared to short-spacing modes. This observation supports the findings of Blake et al. (2014) concerning the structural and physical characteristics of plant wax crystal networks and their link to oil binding capacities [24]. This observation is also consistent with Yao et al. (2021), who investigated the effects of cooling rates on the microstructure and macroscopic properties of rice bran wax oleogels [25]. The phenomenon of anisotropic growth occurs due to variations in molecular structure, composition, and chemical interactions that determine the different rates of fat crystal growth, especially concerning the orientation of the hydrocarbon chain plane.

The XRD graph presented in Figure 6 reveals a single broad maximum within the range of 25° to 35°, indicating the presence of amorphous structures. Notably, there is a complete lack of sharp peaks typically associated with well-organized crystalline structures. This finding is consistent with the composition of the cream [13]. The water content significantly affects the intensity of absorption. Furthermore, Carbopol Aqua SF-1 is an amorphous cross-linked polymer that produces a broad halo without any crystalline diffraction peaks. The only component that might yield signals when subjected to X-ray radiation is bioglass, though it would likely result in low-intensity signals. However, its low concentration of 2% in the mixture and dispersion within a hydrogel matrix effectively obscure any weak reflections. While retinol and vitamin A can exist in a crystalline form, their high water content and solubilization capacity contribute to their molecular dispersion, explaining the absence of characteristic peaks for these compounds.

### 2.4. Electrical Conductivity and pH

The electrical conductivity of the three samples is depicted in Figure 7. The reverse emulsion has the lowest electrical conductivity value at 2.4 µS/cm because the continuous phase is non-polar and contains the weakly polar emulsifier SPAN 60, and water and TEA are found in the discontinuous phase. The direct emulsion has a higher electrical conductivity than the reverse emulsion (353 µS/cm) due to the presence of the emulsifiers with high polarity in the continuous phase—TWEEN 80 and carboxymethyl cellulose (CMC), respectively. It is obvious that the hydrogel demonstrates significantly higher conductivity compared to the two emulsions, attributed principally to Carbopol^®^ Aqua SF-1 OS polymer, a compound that ionizes in the presence of water, and the electrical conductivity of ionic compounds is much higher than that of polar compounds.

As shown in Figure 8, the hydrogel exhibits a higher pH value compared to both the direct and reverse emulsions. This can be attributed to its higher concentration of polyacrylate, specifically the Carbopol^®^ Aqua SF-1 OS polymer, with this one being an acrylate copolymer known to swell under basic conditions. In the direct emulsion, the elevated pH is influenced by a higher triethanolamine (TEA) content. In contrast, the reverse emulsion shows the lowest pH value, likely due to its comparatively lower TEA content.

### 2.5. Stability Tests

For all three samples, we examined the variation in backscattering relative to the height of the container holding the analyzed sample, as well as the changes observed within 6 h of monitoring, measuring every hour. Subsequently, after 30 days of storage at ambient temperature, the backscattering curves were recorded over a period of 4 days.

Across all samples, sedimentation was noted in the lower part of the container, with low backscattering values indicating the presence of sediment. At container heights exceeding 10%, backscattering exhibited a marked increase and tended to stabilize throughout the container’s height. The maximum backscattering value for the hydrogel was approximately 23%, indicating a low concentration of particles within the sample (refer to Figure 9). For container heights greater than 10 mm, the backscattering curve showed variations within a narrow range of less than 2%, suggesting a relatively high stability of the hydrogel-based dispersion. Additionally, a slight decrease in backscattering over the 6 h period was observed, amounting to less than 2%. This decline in signal indicates a degree of instability in the hydrogel-based dispersed system.

The maximum backscattering value for the direct emulsion is approximately 80%, indicating a high concentration of well-dispersed droplets within the sample (refer to Figure 10). The backscattering curve at container height values exceeding 10 mm demonstrates a broader range of backscattering variation compared to the hydrogel, with a difference greater than 5%. This suggests a dynamic regime of coalescence and redispersion of the non-polar phase droplets. Over a period of 6 h, a minimal decrease in backscattering is observed, remaining below 1%. This consistent signal indicates that the direct emulsion exhibits superior stability compared to the dispersed system based on the hydrogel.

The reverse emulsion exhibits lower backscattering than the direct emulsion (Figure 11), with a maximum value of 60%, attributed to a reduced concentration of dispersed particles. The dispersed particles are larger in size, which negatively affects the maximum backscattering value. Additionally, the decrease in backscattering observed over a period of six hours is minimal, remaining below 1%. Overall, this emulsion demonstrates less stability compared to the direct emulsion. Furthermore, the XRD analysis of the reverse emulsion aligns with the observed stability characteristics of this emulsion system. The data indicate that the dispersed particle sizes are comparatively larger, which adversely influences the stability of the RE emulsion.

For the stability measured after 30 days, the hydrogel remains stable. The emulsion samples, especially the reverse emulsion samples, are less stable, with the decrease in backscattering observed over a period of four days having the highest value, as can be seen in Figure 11b.

### 2.6. Stability Tests by Centrifugation

According to the results presented in Table 1, after assessing the stability of the creams at a temperature of 50 °C under the action of centrifugal force, it was observed that after 6 h of analysis time, the sample with the best stability is the hydrogel. This formulation did not show any phase separation during centrifugation, a behavior also supported by the backscattering curve.

In contrast, the reverse emulsion (RE) proved to be the least stable. Thus, phase separation became more pronounced after the first three hours of centrifugation, and after six hours, the separation of the two phases was almost complete. The low stability of the reverse emulsion is due to the low emulsifying power of the sorbitan oleate emulsifier.

A tendency towards phase separation was also observed in the case of the direct emulsion (DE), but this was much lower than in the case of the reverse emulsion. The high stability of the direct emulsion is due to the use of a mixture of two different types of hydrophilic emulsifiers, namely an anionic one (CMC) and a non-ionic one (TWEEN 80), with a synergistic effect towards stabilizing the emulsion.

### 2.7. Microbiological Analyses

All samples were conditioned using a 1/10 dilution by adding sterile physiological serum. Following the dilutions, the culture media were incubated at 37 °C for 24 h, except for Candida albicans, which was incubated for 48 h at the same temperature (Figure 12). After the incubation period, each culture medium was evaluated macroscopically to detect any potential microbial contaminants (Figure 13).

The results showed that there was no development of microbial colonies on any of the culture media tested, suggesting that the evaluated products exhibit microbiological purity (refer to Table 2).

### 2.8. Rheological Analysis

The rheological study of the three formulations was carried out at rotation speeds of 5, 10, and 20 rpm. The spindle selection was made so that the torque had optimal values (preferably between 20 and 80%). Thus, for the hydrogel (Figure 14), spindle v 9.4 was selected, and for the two types of emulsions, spindle v 6.7 and v 9.4 were selected. For the rheological study of the hydrogel, tests were carried out at three temperatures, 20 °C, 25 °C, and 30 °C, respectively, and for the rheological study of the two emulsions, the tests were carried out at 20 C. From Figure 14, it is observed that the viscosity decreases with increasing rotation speed and increasing temperature, which highlights pseudoplastic behavior. Optimal values of the hydrogel viscosity correspond to a speed of 20 rpm at 30 °C, 10 rpm at 25 °C, and 5 rpm at 20 °C, respectively. Thus, the viscosity of the hydrogel at 20 °C and a torque of 51.3% was 65,664 cP. The rheological study of the inverse emulsion was carried out at a temperature of 20 C (Figure 15). This analysis revealed a more pronounced decrease in viscosity with rotation speed for v 9.4 than for spindle 6.7, but more optimal values of torque for spindle v 6.7 both at speeds of 5 rpm (viscosity of 2867 cP) and at speeds of 10 rpm (viscosity of 1939 cP). The rheological study of the direct emulsion was carried out at a temperature of 20 °C (Figure 16) and revealed lower viscosity values compared to the hydrogel and the inverse emulsion. A more pronounced decrease in viscosity with rotation speed is observed for the 9.4 spindle than for the 9.4 spindle.

The study investigates the characteristics of cosmeceuticals formulations, specifically direct emulsions, inverse emulsions, and hydrogels, which synergistically incorporate bioglass and retinol. The objective is to mitigate the irritant potential of retinoids while prolonging their therapeutic efficacy for applications within the health and cosmetics industries. A multifaceted approach employing various characterization methods has been utilized to analyze these formulations comprehensively. Analytical techniques, including thermogravimetric analysis (TGA), differential scanning calorimetry (DSC), and X-ray diffraction (XRD), were employed alongside stability and microbiological assessments. The results indicate that the electrical conductivity of the formulations is influenced by both water content and the polarity of the continuous phase. Notably, the hydrogel exhibited markedly higher conductivity (76% water content) compared to the two emulsions, with the inverse emulsion displaying the lowest conductivity value at 2.4 µS/cm, attributed to the non-polar nature of its continuous phase. The pH values of the formulations were determined by the triethanolamine (TEA) content present in the continuous phase. TGA and DSC analyses suggested a significant shift in the temperature at which maximum water loss occurred, indicative of stronger interactions between water molecules and the organic compounds present, particularly in the emulsions as opposed to the hydrogel. Fourier-transform infrared (FTIR) spectroscopy confirmed the presence of Si-O-Ca and Si-O-Si bonds, consistent with the incorporation of bioglass. XRD analysis was conducted to evaluate the crystallinity of the prepared materials, allowing for a quantitative assessment of any crystalline phases present, including those associated with the bioglass or organic compounds with high melting points. A diffraction maximum in the range of 25° to 35° was linked to hydroxyapatite reflections. Additionally, the absence of diffraction peaks corresponding to oxide structures suggests a completely amorphous structure of the utilized bioglass. Stability analyses, assessed via backscattering measurements, revealed differing behaviors between the hydrogel and the emulsions. The hydrogel demonstrated a lower backscattering value compared to the emulsions, indicating a reduced concentration of dispersed particles. Furthermore, significant differences were recorded between the two emulsion types, with the direct emulsion exhibiting a higher backscattering value and consequently a greater droplet content than the inverse emulsion. Both the emulsions and the hydrogel demonstrated high stability over a period of up to 30 days, evidenced by minimal variations in backdiffusion, but centrifugation stability tests, performed at a temperature of 50 °C, highlighted the superior stability of the hydrogel. Further investigation by microstructural imaging methods was not possible due to the continuous fluctuation of the droplet diameter after dilution, and fluctuations were observed in preliminary dynamic light scattering (DLS) tests. This behavior is attributed to the complex composition of these creams, which includes the solubilization properties of hydrophilic compounds in the lipophilic phase by polyethoxylated sorbitan esters when subjected to high dilutions. Microbiological analyses confirmed the absence of microbial colonization on any tested culture medium, indicating that all formulations are microbiologically pure.

## 3. Conclusions

Attenuation of the irritant potential of retinol, while maintaining its therapeutic efficacy, can be achieved by formulating it in combination with bioglass. This study introduces three distinct variants for retinol conditioning: direct emulsion, inverse emulsion, and hydrogel. To evaluate the effectiveness of these formulations, a wide range of analytical techniques was used, including thermogravimetric analysis, differential scanning calorimetry, X-ray diffraction, stability assessments, and rheological and microbiological studies. The aim of these comprehensive analyses was to select the optimal variant for cosmeceuticals applications based on the performances of the three types of formulations.

Thus, the high viscosity of the three formulation variants ensures high protection against contamination with microbial agents. In addition, the low value of back diffusion in the case of the hydrogel suggests a lower particle concentration, which could promote a slower release of retinol. This characteristic, associated with the high stability evidenced by centrifugation and the higher viscosity than the other two types of formulations, makes the hydrogel formulation the optimal option for the preparation of retinol-based creams for dermal administration.

## 4. Materials and Methods

### 4.1. Materials and Reagents

To prepare the emulsions, several products were used: retinol (Merck, Darmstadt, Germany, ≥95.0%), Bioglass^®^ powder Schott (Schott UK Ltd., Wolverhampton, UK), lanolin (USP, Sigma-Aldrich, St. Louis, MO, USA), hyaluronic acid-HA (Oligo-HA4, Sigma-Aldrich), polymer-TEA (triethanolamine, Merck, reagent grade, 97%), and sodium carboxymethyl cellulose-CMC (Sigma-Aldrich). Sabosorb MO-Span 60 (sorbitan monostearate) (Sabo) was utilized for stabilizing the reverse emulsion (RE), while Tween 80 (polyoxyethylenesorbitan monooleate) (Sigma-Aldrich) was employed to stabilize the direct emulsion (DE). Additionally, polyacrylic acid (Carbopol Aqua SF-1 OS polymer) (Lubrizol) was used in the formulation of the hydrogel. For the non-polar phase, coconut and soybean oils (Solaris Plant) and ultra-pure water and WIFI-quality water (Pixico, Bucharest, Romania) were used.

### 4.2. Synthesis of Creams

A direct emulsion, an inverse emulsion, and a hydrogel were prepared through mechanical stirring using an IKA 18 ULTRA-TURRAX digital disperser (IKA-Werke GmbH & Co. KG, Staufen, Germany) at a speed of 9000 rpm and a temperature of 40 °C for a duration of 30 min. The selection of components for the three formulations carried out taking into account their specific characteristics for use in the dermatological field. Thus, in the preparation of the direct emulsion, we used two emulsifiers, one hydrophilic nonionic (Tween 80) and one anionic (CMC). In the preparation of the reverse emulsion, we used a hydrophobic nonionic emulsifier (Span 60). The stabilization of the hydrogel was carried out in the presence of an acrylic copolymer in the Na form (Carbopol Aqua SF-1 OS).

The composition and concentration of the components in the resulting creams are outlined in Table 3.

### 4.3. Characterization Methods

The following analyses were performed for the creams obtained in the study: thermogravimetric analysis (TGA), differential scanning calorimetry (DSC*),* Fourier transform infrared spectroscopy (FTIR), X-ray diffraction (XRD), electrical conductivity, pH, stability analyses, and microbiological analyses.

Thermogravimetric analysis for DE, RE and hydrogel were performed with the Thermal Analysis System TGA 2 apparatus from METTLER TOLEDO (Greifensee, Switzerland), in the 25–700 °C temperature range, in a nitrogen atmosphere, with a heating rate of 10 °C/min.

Differential scanning calorimetry was performed to investigate temperature-induced transitions. The analysis was conducted using a Thermal Analysis System DSC 3+ from Mettler Toledo (Greifensee, Switzerland). The samples were heated from room temperature to 400 °C at a rate of 10 °C per minute in a nitrogen atmosphere.

For the qualitative analysis of the materials, Fourier Transform Infrared was used to identify the functional groups present in the structures. The analysis was conducted using a Shimadzu IRAffinity-1S spectrophotometer (Kyoto, Japan), which was equipped with the GladiATR-10 accessory. The measurements were taken within the wavelength range of 380 to 4000 cm^−1^, with a spectral resolution of 4 cm^−1^.

Determination of the conductivity and pH of the creams was performed with the inoLab Multi 9630 IDS (WTW, Berlin, Germany) multimeter. The electrode used for pH is SeTix 980 with glass rod and temperature sensor. The conductivity cell was WTW IDS TetraCon 925.

The degree of crystallinity of the prepared materials was determined by X-ray diffraction performed with a Bruker X-ray diffractometer (Bruker-AXS, Karlsruhe, Germany) equipped with a Cu-Kα source at 40 kV and 5 mA, 2θ range 1–40 at a rate of 10° min^−1^.

The stability of the creams was evaluated both by determining the tendency to sedimentation, coalescence, or aggregation and by centrifugation. Thus, using the Turbiscan Lab Expert Formulation (Toulouse, France), we measured the variation in backscattering over time as a measure of the stability of these creams. Centrifugation tests were performed on a ROTOFIX 46 centrifuge (Târgoviște, Romania) at 50 °C and 1500 rpm, for a duration of up to 6 h.

For microbiological analyses, contamination with 6 types of microorganisms was performed to assess if encapsulation promoted their growth. A 1/10 dilution was made from each media and homogenized using a sterile loop. 0.1 mL of the diluted sample was pipetted onto the Petri plate with the culture medium specific to the desired analysis. The handling of laboratory equipment during cosmetic product testing (weighing, mixing, and pouring into Petri dishes) was designed to avoid sample contamination [26,27]. The sample was dispersed uniformly by rotating the plate, previously noted with the name of the medium [28]. The culture media used were [29] 

Nutrient Agar (NA) medium, which allows the cultivation and isolation of aerobic bacteria from various sources. On this type of media, we can observe and characterize the morphology of microbial colonies, pigmentation and other physical characteristics of them.Mannitol Salt Agar (MSA) medium, which is a selective medium with a high salt concentration, which serves in the isolation, identification and differentiation of staphylococci, especially Staphylococcus aureus.Cetrimide medium (CET), which is used for the isolation and identification of Pseudomonas bacteria, especially Pseudomonas aeruginosa, a pathogenic bacterium associated with various infections.CLED-Agar medium, which is a specialized culture medium, to detect the growth of Gram-negative bacteria, such as Enterobacter species, Escherichia coli and others.MacConkey Agar, which is used for the isolation of Gram-negative enteric bacteria, especially Escherichia coli and coliform bacteria, which are commonly found in the intestinal tract and can be indicators of fecal contamination.Sabourand-Agar, which is used for the isolation of fungi, especially yeasts and molds.

Rheological studies were performed using an IKA ROTAVISC me-vi viscometer (Ika, Staufen, Germany). Each sample was measured three times, at three rotation speeds: 5, 10 and 20 rotations per minute. For the hydrogel, the tests were performed at temperatures of 20 °C, 25 °C and 30 °C, respectively, and for the direct and reverse emulsions, the measurement was performed at a temperature of 20 °C.

### 4.4. Statistical Analysis

Both the pH and electrical conductivity stability were evaluated through three consecutive measurements for each type of cream. The results were subsequently used to calculate experimental errors using the standard deviation statistical function, which provides insight into the dispersion of the data around the mean values. To assess whether the standard deviation is high or low, indicating that the values are either widely varied or closely clustered around the mean, the coefficient of variation was employed.

## Figures and Tables

**Figure 1 gels-11-00935-f001:**
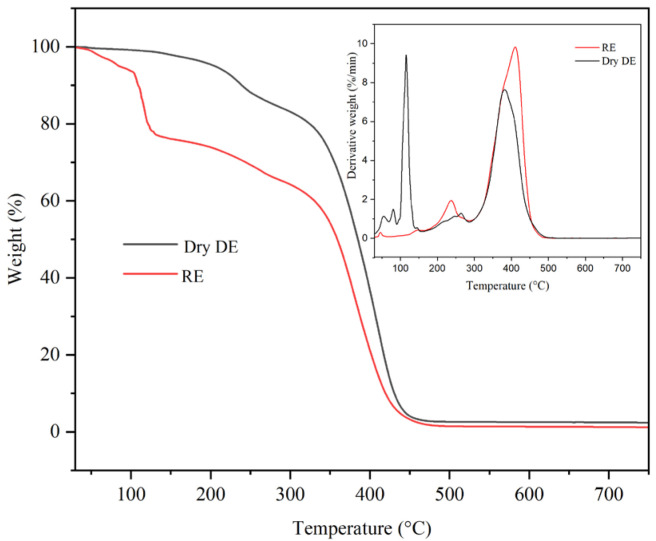
Thermogravimetric analysis of the two emulsions.

**Figure 2 gels-11-00935-f002:**
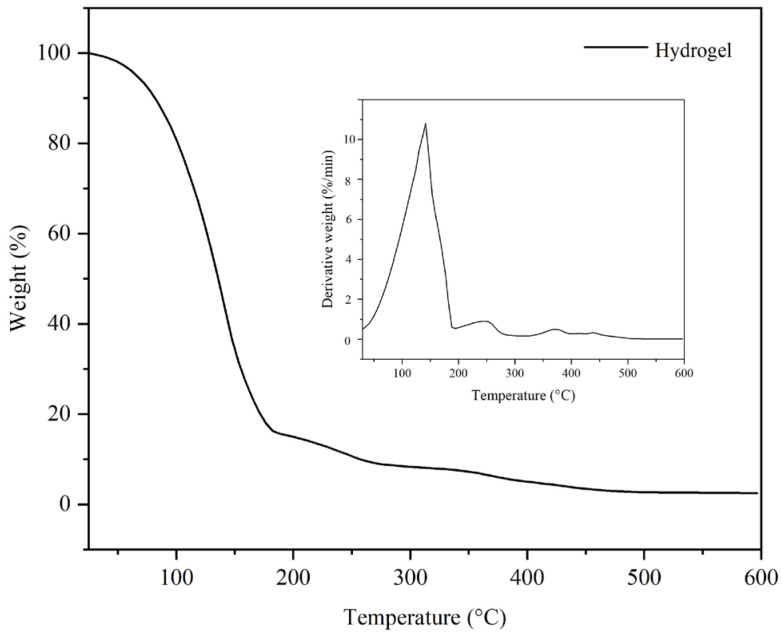
Thermogravimetric analysis of the hydrogel.

**Figure 3 gels-11-00935-f003:**
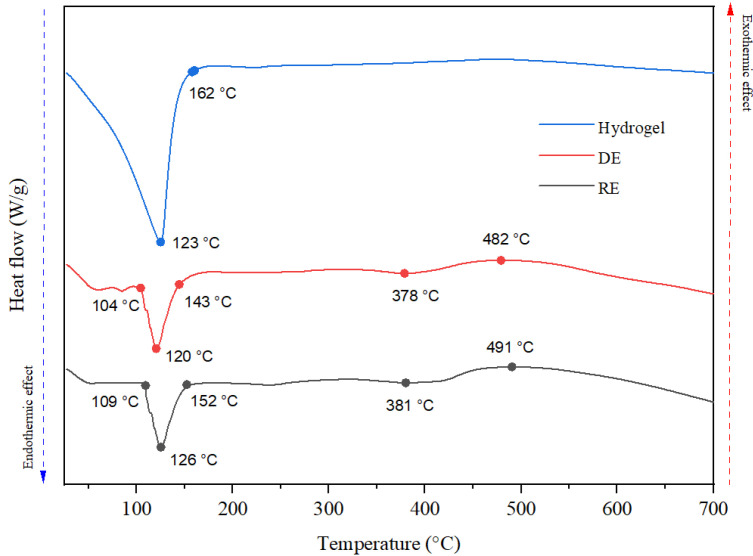
DSC thermogram of the hydrogel and the two emulsions.

**Figure 4 gels-11-00935-f004:**
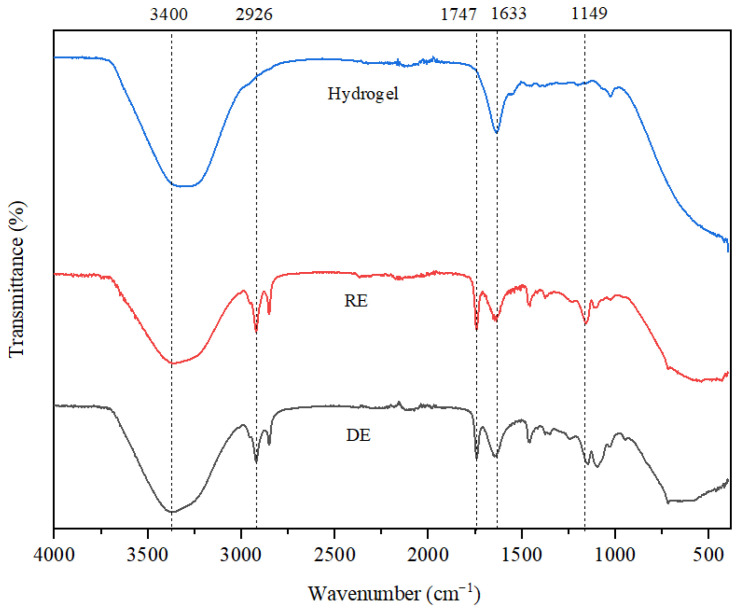
The FTIR spectra of the hydrogel and the two emulsions.

**Figure 5 gels-11-00935-f005:**
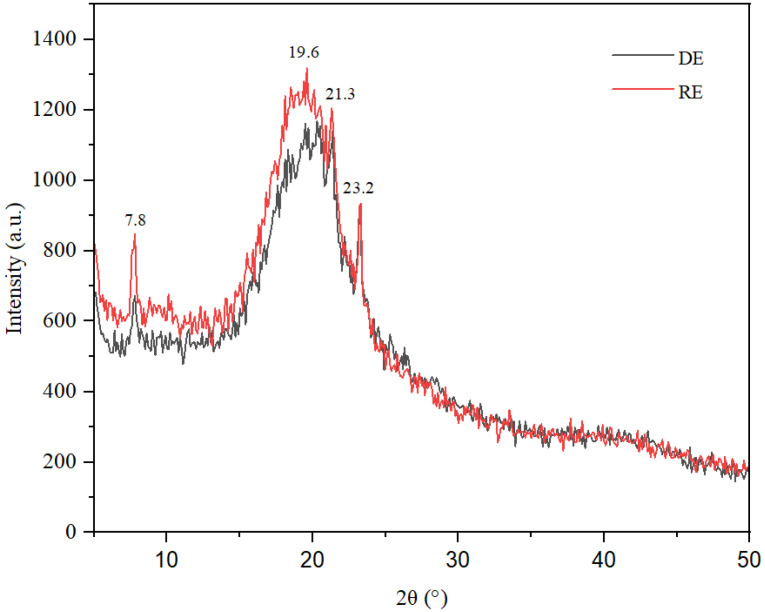
XRD spectra of the two emulsions.

**Figure 6 gels-11-00935-f006:**
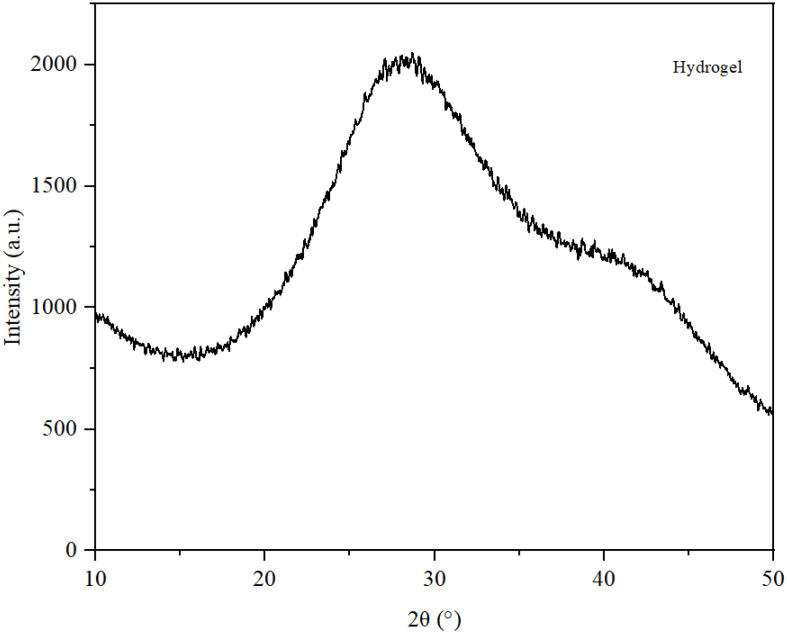
XRD spectra of the hydrogel.

**Figure 7 gels-11-00935-f007:**
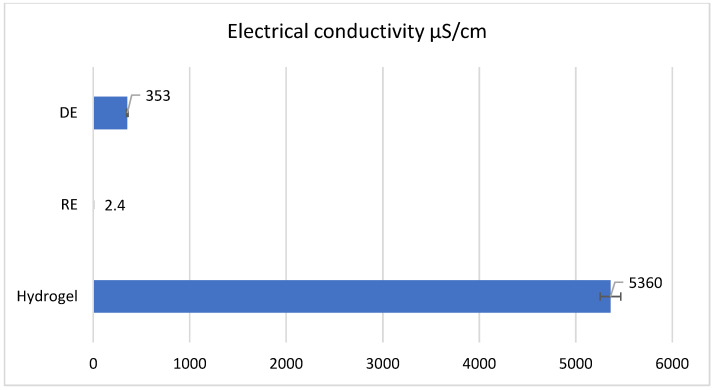
The conductivity of the samples.

**Figure 8 gels-11-00935-f008:**
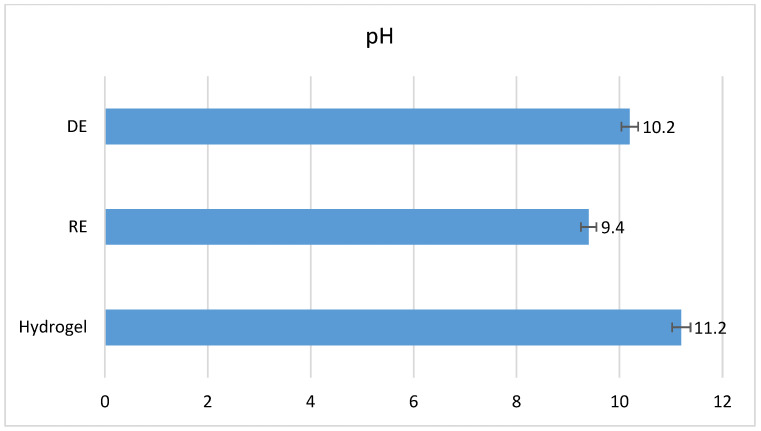
The pH of the samples.

**Figure 9 gels-11-00935-f009:**
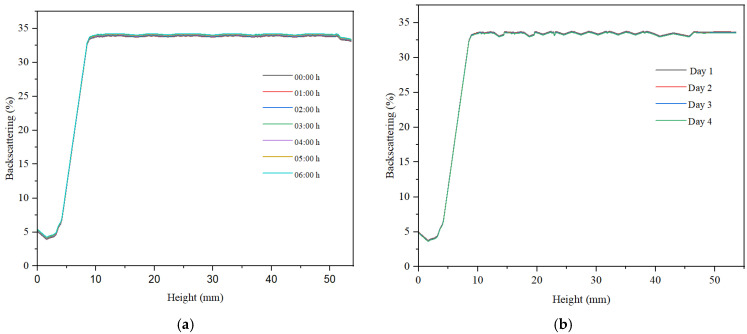
Backscattering curve for hydrogel (**a**) at preparation and (**b**) after 30 days.

**Figure 10 gels-11-00935-f010:**
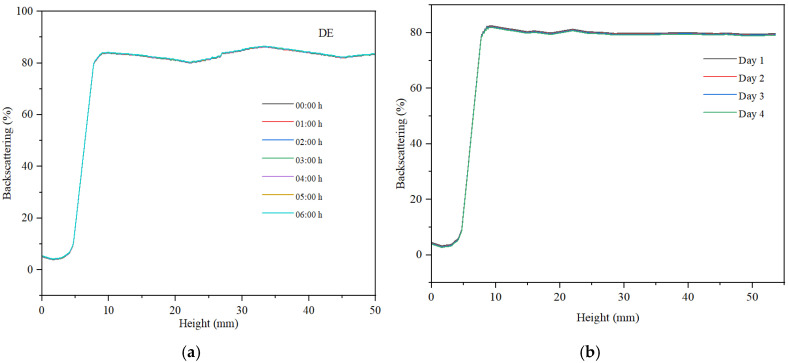
Backscattering curve for direct emulsion (**a**) at preparation, (**b**) after 30 days.

**Figure 11 gels-11-00935-f011:**
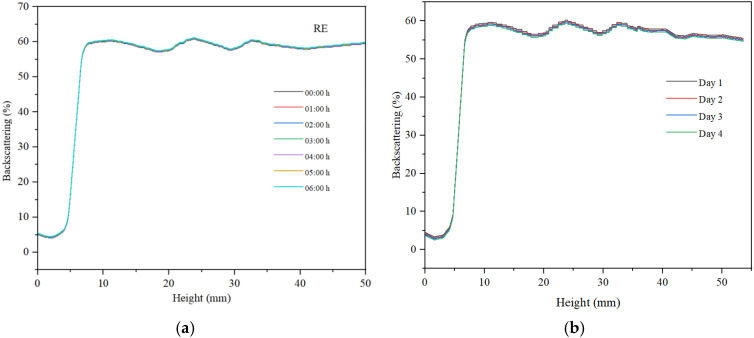
Backscattering curve for reverse emulsion (**a**) at preparation and (**b**) after 30 days.

**Figure 12 gels-11-00935-f012:**
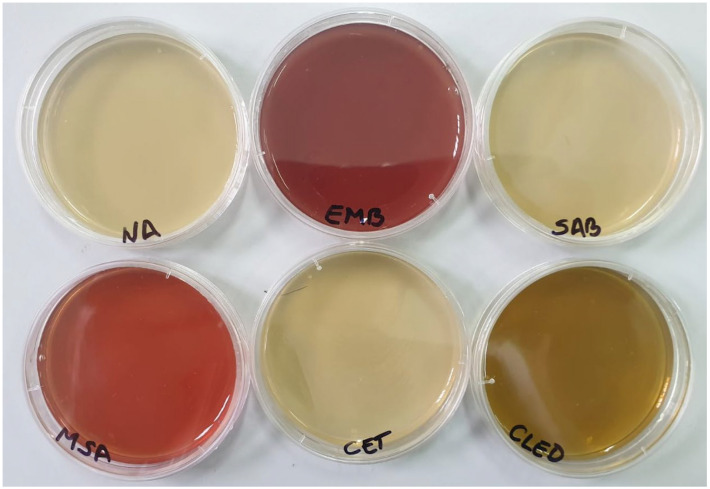
Culture media inoculated with the cosmetic product before incubation.

**Figure 13 gels-11-00935-f013:**
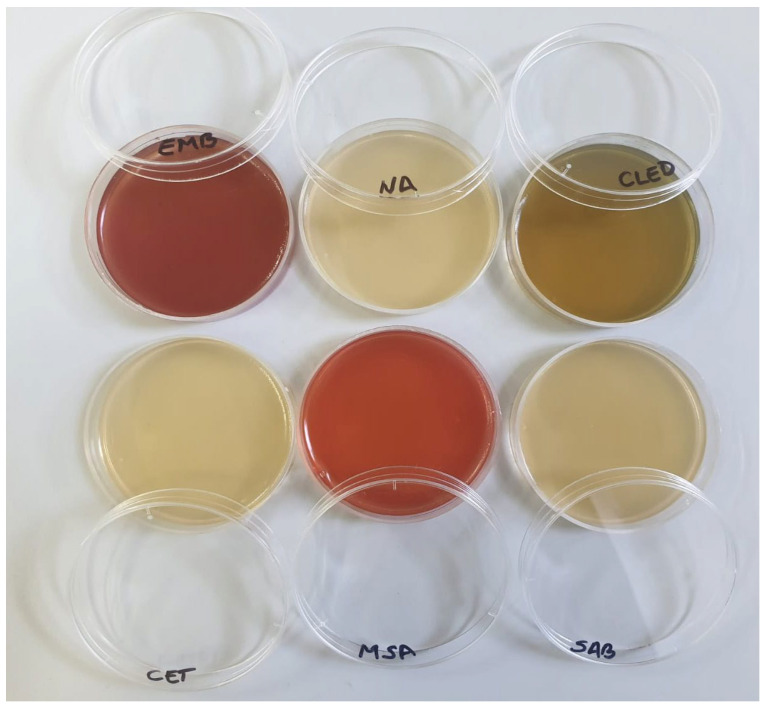
Culture media inoculated with the cosmetic product after incubation for the three types of creams.

**Figure 14 gels-11-00935-f014:**
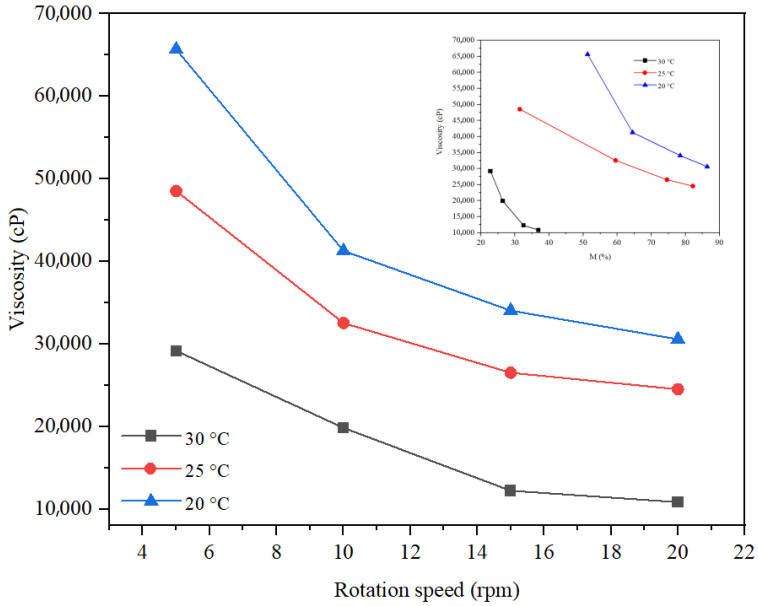
Rheological behavior of hydrogel.

**Figure 15 gels-11-00935-f015:**
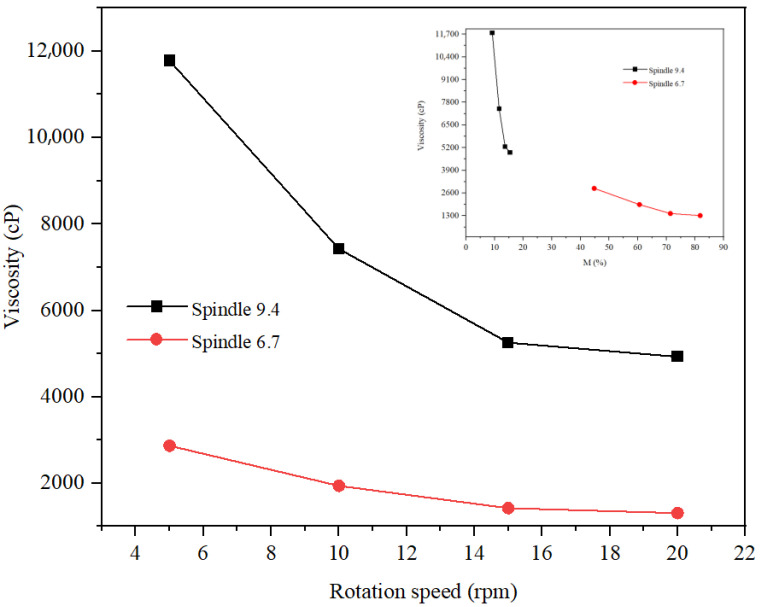
Rheological behavior of the RE.

**Figure 16 gels-11-00935-f016:**
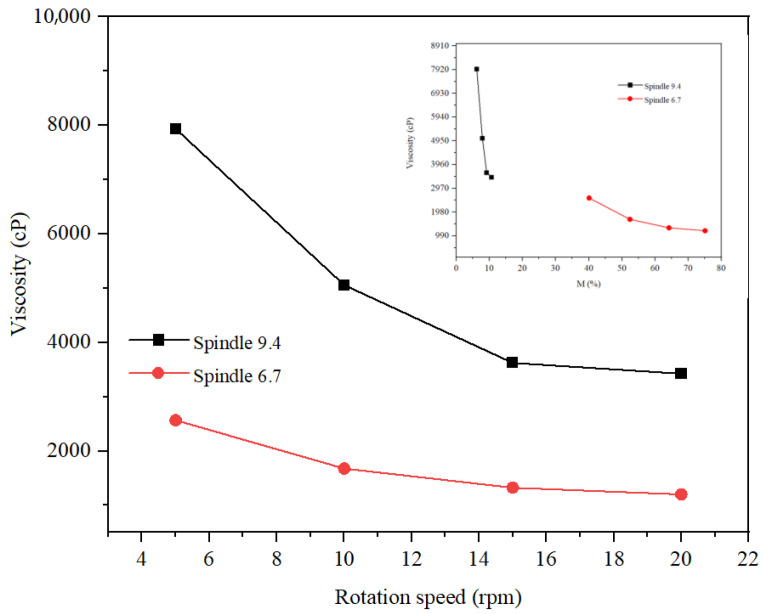
Rheological behavior of the DE.

**Table 1 gels-11-00935-t001:** Stability analysis during centrifugation.

Sample	Upper Phase After 1 h (%*v*/*v*)	Upper Phase After 2 h (%*v*/*v*)	Upper Phase After 3 h (%*v*/*v*)	Upper Phase After 4 h (%*v*/*v*)	Upper Phase After 5 h (%*v*/*v*)	Upper Phase After 6 h (%*v*/*v*)
*Hydrogel*	0	0	0	0	0	0
DE	0.2	0.45	0.8	1.9	3.1	3.4
RE	0.9	7.4	17.7	29.3	42.1	44.7

**Table 2 gels-11-00935-t002:** Microbiological analyses and culture media used.

Nr crt.	Name Analysis	Culture Media	Codification	Reverse Emulsion	Direct Emulsion	Hydrogel
1	Total aerobic bacteria	Nutrient Agar medium	NA	Absent	Absent	Absent
2	Pathogenic bacteria-*E. coli*	MacConkey Agar medium	EMB	Absent	Absent	Absent
3	Pathogenic bacteria-*Staphylococcus aureus*	MSA medium	MSA	Absent	Absent	Absent
4	Pathogenic bacteria-*Pseudomonas aeruginosa*	Cetrimide medium	CET	Absent	Absent	Absent
5	Pathogenic bacteria-*Enterobacter sp.*	CLED medium	CLED	Absent	Absent	Absent
6	Pathogenic fungi-*Candida albicans*	Sabouraud Chloramphenicol Agar Medium	SAB	Absent	Absent	Absent

**Table 3 gels-11-00935-t003:** Contents of the creams.

Component	Reverse Emulsion-RE, (%wt.)	Direct Emulsion-DE, (%wt.)	Hydrogel (%wt.)
Ultrapure water	40.00	40.00	76.00
Coconut oil	25.00	25.00	-
Soybean oil	11.00	10.00	-
Lanolin	15.50	13.50	-
TEA	1.50	2.75	-
Retinol	2.00	2.00	3.00
Bioglass	1.25	1.25	2.00
Hyaluronic acid	0.50	0.50	1.50
CMC	-	0.50	-
Span 60	3. 25	-	-
Tween 80	-	4.50	-
Carbopol Aqua SF-1 OS	-	-	15
Vitamine E	-	-	1.00
Cetyl alcohol	-	-	1.5

## Data Availability

The original contributions presented in this study are included in the article. Further inquiries can be directed to the corresponding author.

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
