# Peer review of "Influence of Retinol Dermal Delivery Formulation on Its Stability Characteristics"

_gels, 2025, doi:10.3390/gels11120935_

Round 1
Reviewer 1 Report
Comments and Suggestions for Authors
The article, titled "The Effect of Retinol Delivery Formula on Retinol Stability," addresses an interesting and important topic related to cosmetic formulation science and the stabilization of retinol using gels and emulsions containing bioglass. This concept falls within the scope of the journal "Żele" (Gels) and has potential scientific and practical importance. However, in its current form, the study lacks sufficient methodological depth and analytical rigor to justify publication in this journal. Only basic physicochemical analyses were performed (TGA, DSC, FT-IR, XRD, pH, conductivity, and short-term stability). These techniques are standard and do not provide sufficient insight into the gel structure, rheological properties, or controlled retinol release. Therefore, comprehensive rheological analysis (viscoelastic parameters, viscosity profiles) and microstructural imaging (e.g., SEM, TEM, or confocal microscopy) are strongly recommended. Stability was assessed for only 6 hours, which is clearly too short a period to assess the long-term stability of emulsions and hydrogels intended for cosmetic use. Accelerated aging tests (temperature cycling, centrifugation, and observations lasting several weeks or months) should be included. Because the main assumption concerns "delivery" and therapeutic efficacy, quantitative data on retinol release kinetics or diffusion would be essential to support conclusions. Interpretation of the thermal and spectroscopic data is primarily descriptive. A more analytical discussion (including numerical data, comparative analysis, and error ranges) would strengthen the manuscript.
Comments on the Quality of English LanguageThe language is correct, but at times imprecise and too descriptive.
Author Response
Reviewer 1
Thank you very much for taking the time to review this manuscript.
- However, in its current form, the study lacks sufficient methodological depth and analytical rigor to justify publication in this journal. Only basic physicochemical analyses were performed (TGA, DSC, FT-IR, XRD, pH, conductivity, and short-term stability). These techniques are standard and do not provide sufficient insight into the gel structure, rheological properties, or controlled retinol release. Therefore, comprehensive rheological analysis (viscoelastic parameters, viscosity profiles) and microstructural imaging (e.g., SEM, TEM, or confocal microscopy) are strongly recommended.
Answer:
We have completed the work with a rheological study.
The microstructural imaging methods proposed in the specialized literature (SEM, TEM, or confocal microscopy, etc.) assume the dilution of the respective formulations, dilution which in the case of our recipes modifies the size of the particles or droplets and implicitly the granulometric distribution, providing erroneous information about these creams. The preliminary dynamic light scattering (DLS) analysis tests that we performed highlighted a continuous fluctuation of the particle diameter after dilution, making it impossible to complete the analysis. This behavior is probably due to the complex composition of these creams but also to the characteristics of the emulsifiers used. For example, Tween 80 at low concentrations is a solubilizing agent for hydrophilic compounds in the lipophilic phase.
- Stability was assessed for only 6 hours, which is clearly too short a period to assess the long-term stability of emulsions and hydrogels intended for cosmetic use. Accelerated aging tests (temperature cycling, centrifugation, and observations lasting several weeks or months) should be included.
Answer:
We completed the paper with additional data on long-term stability including accelerated aging tests by centrifugation at 50°C.
- Because the main assumption concerns "delivery" and therapeutic efficacy, quantitative data on retinol release kinetics or diffusion would be essential to support conclusions. Interpretation of the thermal and spectroscopic data is primarily descriptive. A more analytical discussion (including numerical data, comparative analysis, and error ranges) would strengthen the manuscript.
Answer:
In this study, we comparatively evaluated the three variants of retinol-based formulations for dermal administration, namely direct emulsion, reverse emulsion, and hydrogel, to identify the most stable variant. Based on this study, we selected the most stable formulation, improved its composition, and studied the therapeutic efficacy by testing it on the skin. The results are presented in the paper „Retinol Dispersion In The Form Of Hydrogel For Dermal Delivery”, STUDIA UBB CHEMIA, LXX, 2, 2025 (p. 37-51), DOI:10.24193/subbchem.2025.2.03.
We have completed with numerical data regarding comparative analysis and error ranges.

Reviewer 2 Report
Comments and Suggestions for Authors
Please find attached.

Author Response
Reviewer 2
Thank you very much for taking the time to review this manuscript.
- The details on preparation and differences in the composition that lead to emulsion, reverse emulsion and hydrogel and the reasons why this happens should be discussed. The ways of confirming and distinguishing between these three formulations should be discussed.
Answer:
The issue was solved.
- The source of the used “recipes” or how they were determined should be mentioned.
Answer:
We proposed new recipes for obtaining emulsions and hydrogels by using bioemulsifiers. To formulate the recipes, we used two phases with different polarities: water and a lipid phase containing coconut oil, soybean oil, and lanolin. The proportion of the two phases was adjusted. In the presence of dedicated emulsifiers — SPAN 60 for reverse emulsion and TWEEN 80 or carboxymethyl cellulose (CMC) for direct emulsion — the best possible stability will be ensured.
The recipes presented in the study are the result of optimizing the proportions of the cream components.
- Thermal properties. The differences between the three formulations should be discussed in relation their different structure and not only composition.
Answer:
The issue was solved.
- The peaks of DE and RE that are missing from the hydrogel should be explained.
Answer:
In the RE and DE emulsions the vibrations at 1747 cm⁻¹ correspond to C=C bonds within the structure of unsaturated triglycerides, while those at 1149 cm⁻¹ highlight the ester bonds typical of triglycerides. These signals are only highlighted in the case of the two emulsions, as the hydrogel does not contain coconut oil, soybean oil or lanolin.
- The sharp peaks that are discussed should be shown in the graph (as for example in FTIR).
Answer:
The issue was solved.
- Fig 4. Indication of the different spectra (hydrogel, DE, RE) are missing.
Answer:
The issue was solved.
- Line 373-375. Mention which ions exactly cause the conductivity.
Answer:
The issue was solved
- Explain (based on formulation structure and composition) the differences found and discussed on stability (backscattering results).
Answer:
Backscattering results were complemented with additional stability tests after 30 days. We also performed stability evaluation tests by centrifugation.
- Antimicrobial activity. Explain why the formulations remain pure.
Answer:
The purity of the formulations is attributed to the encapsulation method employed in the three formulation types and their high viscosity. These factors protect the active principles from microbial attack.
- Line 181. The phrase “backscattering diffusion” is not proper.
Answer:
The issue was solved.
- In TGA. The authors should present the derivative data as well. They should also revise the discussion as the terms “mass loss” and weight are confused.
Answer:
The issue was solved, the derivative data are presented.
- I think the authors mean “backscattering” when they mention “backdiffusion”.
Answer:
The issue was solved.
- In the Conclusions a general phrase on the usefulness of this work can be added. The result of the antimicrobial activity could also be mentioned here.
Answer:
The issue was solved.
- “Electrical conductance” is not a correct term.
Answer:
The issue was solved. “Electrical conductance” was corrected with “Electrical conductivity”.

Reviewer 3 Report
Comments and Suggestions for Authors
The authors studied the preparation and characterization of composites containing retinol for skincare applications. The manuscript is generally clearly written. The introduction appropriately presented the previous work in the field of cosmetics and the use of the different components of the prepared composites. The experiment is written with sufficient details. The results are well-discussed and interpreted.
I recommend publication after considering the following:
- In the introduction, line 116, the author wrote “ glass with composition 58S”. What is this glass composition?
- In the results, Figure 4 lacks the legends of the curves.
- The pictures of the antimicrobial test should be added either to the text or as a supplementary file.
Author Response
Reviewer 3
Thank you very much for taking the time to review this manuscript.
- In the introduction, line 116, the author wrote “glass with composition 58S”. What is this glass composition?
Answer:
We corrected with “bioactive glass, type 58S”. The composition of 58S bioglass is 58 wt% SiO2, 33 wt% CaO and 9 wt% P2O5, according E. Cañas, A. Borrell, R. Benavente, M.D. Salvador, “Synthesis of 58S bioactive glass based on a novel methodology employing microwave technology”, Ceramics International, 50(13), Part B, 2024,p. 24471-24478.
- In the results, Figure 4 lacks the legends of the curves.
Answer:
The issue was solved.
- The pictures of the antimicrobial test should be added either to the text or as a supplementary file.
Answer:
The issue was solved by adding images before and after antimicrobial test.

Round 2
Reviewer 1 Report
Comments and Suggestions for Authors
The revised version shows clear improvement compared to the original submission. The authors have addressed most of the critical comments and added valuable new data. In particular, the inclusion of rheological analysis and extended stability tests (both accelerated and 30-day observations) has significantly strengthened the methodological part of the paper. The statistical treatment of the results is also appreciated and adds credibility to the data interpretation.
The explanation regarding the difficulty of performing microstructural imaging is reasonable and technically justified. Although direct SEM/TEM analysis would have been helpful, I agree that the provided discussion and the DLS observations sufficiently explain the limitation. It might still be worth mentioning this explicitly as a limitation in the manuscript.
Overall, the work is now much more complete and scientifically sound. The topic is relevant for the field of cosmetic formulation and controlled delivery systems.
I Accept after minor revision (mainly minor language polishing and small clarifications).
Comments on the Quality of English LanguageThe language is correct, but at times imprecise and too descriptive.
Author Response
The explanation regarding the difficulty of performing microstructural imaging is reasonable and technically justified. Although direct SEM/TEM analysis would have been helpful, I agree that the provided discussion and the DLS observations sufficiently explain the limitation. It might still be worth mentioning this explicitly as a limitation in the manuscript.
Overall, the work is now much more complete and scientifically sound. The topic is relevant for the field of cosmetic formulation and controlled delivery systems.
I Accept after minor revision (mainly minor language polishing and small clarifications).
Answer:
We have revised the language and introduced further clarifications.
Both the emulsions and the hydrogel demonstrated high stability over a period of up to 30 days, evidenced by minimal variations in backdiffusion, but centrifugation stability tests, performed at a temperature of 50 °C, highlighted the superior stability of the hydrogel.
Further investigation by microstructural imaging methods was not possible due to the continuous fluctuation of the droplet diameter after dilution, fluctuations observed in preliminary dynamic light scattering (DLS) tests. This behavior is attributed to the complex composition of these creams, which includes the solubilization properties of hydrophilic compounds in the lipophilic phase by polyethoxylated sorbitan esters when subjected to high dilutions.

Reviewer 2 Report
Comments and Suggestions for Authors
The authors solved all issues. It would be helpful if they had included the exact changes they made in the manuscript within their answer too.
Author Response
The authors solved all issues. It would be helpful if they had included the exact changes they made in the manuscript within their answer too.
Answer:
We have revised the language. Thank you for your suggestion, these considerations will be incorporated into future work.

Reviewer 3 Report
Comments and Suggestions for Authors
The authors appropriately responded to the reviewer comments.
Author Response
The authors appropriately responded to the reviewer comments.
Answer:
Thank you for your suggestions regarding our paper.
